# Differentiable Neural Architecture Search in Equivalent Space with Exploration Enhancement

**Miao Zhang**[1,2,3]**, Huiqi Li**[1]***, Shirui Pan**[3]***, Xiaojun Chang**[3]**, Zongyuan Ge**[3,4]**, Steven Su**[2]
[1]Beijing Institute of Technology    [2]University of Technology Sydney
[3]Monash University    [4]Airdoc Research Australia
{Miao.Zhang, Shirui.Pan, Xiaojun.Chang, Zongyuan.Ge}@monash.edu,
huiqili@bit.edu.cn, steven.su@uts.edu.au

## Abstract

Recent works on One-Shot Neural Architecture Search (NAS) mostly adopt a bilevel optimization scheme to alternatively optimize the supernet weights and architecture parameters after relaxing the discrete search space into a differentiable space. However, the non-negligible incongruence in their relaxation methods is hard to guarantee the differentiable optimization in the continuous space is equivalent to the optimization in the discrete space. Differently, this paper utilizes a variational graph autoencoder to injectively transform the discrete architecture space into an equivalently continuous latent space, to resolve the incongruence. A probabilistic exploration enhancement method is accordingly devised to encourage intelligent exploration during the architecture search in the latent space, to avoid local optimal in architecture search. As the catastrophic forgetting in differentiable One-Shot NAS deteriorates supernet predictive ability and makes the bilevel optimization inefficient, this paper further proposes an architecture complementation method to relieve this deficiency. We analyze the proposed method's effectiveness, and a series of experiments have been conducted to compare the proposed method with state-of-the-art One-Shot NAS methods.

## 1 Introduction

While Neural Architecture Search (NAS) [9, 18, 29] has achieved impressive results in many automating neural network designing tasks, it has also imposed huge demand of computation power for most machine learning practitioners. To mitigate this problem, many recent studies have been devoted to reducing the search cost through the weight-sharing paradigm (which is also called One-Shot NAS) [4]. These methods define a supernet to subsume all possible architectures in the search space, and evaluate architectures through inheriting weights from the supernet. Early One-Shot NAS approaches first adopt a controller to sample architectures for the supernet training, and then use heuristic search methods to find the promising architecture over a discrete search space based on the trained supernet [14, 19, 26]. Later researches [6, 11, 15, 22, 23, 32] further employ the continuous relaxation to make the architecture differentiable, so that gradient descent can be used to optimize the architecture with respect to validation accuracy, and this paradigm is also referred to as differentiable NAS [22].

One shortcoming for the discrete-continuous conversion in differentiable NAS is that there is no theoretical foundation showing that the optimization in the continuous latent space is equivalent to discrete space. The lack of injective constraints in the simple continuous relaxation hardly guarantees that performing optimization in the continuous latent space is equivalent to doing so in the discrete space. Several concurrent works [7, 36] further reveal that this *incongruence*, which is correlated

---

with the Hessian norm of architecture parameters, constantly increases during the architecture search of differentiable NAS. In addition, current differentiable NAS methods only rely on the performance reward to update the architecture parameters. This method entails the *rich-get-richer problem* [1, 40], since architectures with better performance in the early stage would be trained more frequently, and the updated weights further make these architectures having a higher probability of being sampled, which easily leads to a local optimal.

Another limitation of differentiable NAS is the *catastrophic forgetting problem* arisen in the training process. Differentiable methods assume that the inner supernet weights learning in each step improves the validation performance of all architectures with inheriting the supernet weights. However, this assumption may not hold. In practice, each step of supernet training in One-Shot NAS usually deteriorates other architectures' validation performance containing partially shared weights with currently learned architecture [5]. This forgetting problem is less studied in differentiable NAS.

Motivated by the aforementioned observations, this paper develops an **E**xploration **E**nhancing **N**eural **A**rchitecture **S**earch with Architecture Complementation (**E²NAS**) to address the limitations faced by existing differentiable NAS approaches. For the *incongruence* in the relaxation transformation of differentiable NAS, we utilize a variational graph autoencoder with an asynchronous message passing scheme to transform the discrete architectures into an equivalent continuous space injectively. Because of the injectiveness, we could equivalently perform optimization in the continuous latent space with a solid theoretical foundation [33, 38]. For the *rich-get-richer problem* entailed by the reward-based gradient methods, we devised a probabilistic exploration enhancement method to encourage intelligent exploration during the architecture search in the latent space. As to the common *catastrophic forgetting* in differentiable NAS, an architecture complementation based continual learning method is further proposed for the supernet training, to force the supernet to keep the memory of previously visited architectures. We compared the proposed approach with different One-Shot NAS baselines on the NAS benchmark dataset NAS-Bench-201 [13], and extensive experimental results illustrate the effectiveness of our method, which outperforms all baselines on this dataset.

## 2 Backgrounds

**Differentiable NAS** is built on One-Shot NAS which encodes the search space $\mathcal{A}$ as an over-parameterized network (supernet) $\mathcal{W}_{\mathcal{A}}$, and all candidate architectures $\alpha$ directly inherit weights $\omega = \mathcal{W}_{\mathcal{A}}(\alpha)$ from the trained supernet for evaluation. One-Shot NAS uses a controller to sample discrete architectures from the search space for supernet training, and the most promising architecture $\alpha^*$ is obtained through heuristic search methods based on the trained supernet. Differentiable NAS [2, 6, 22, 24, 32] further relaxes the discrete architecture into continuous space $\mathcal{A}_{\theta}$, and alternatively learn the architecture parameters and supernet weights based on gradient methods. The best architecture with continuous representation $\alpha_{\theta}$ can be obtained once the supernet training is finished through:

$$\min_{\alpha_{\theta} \in \mathcal{A}_{\theta}} \mathcal{L}_{\text{val}}(\operatorname*{argmin}_{\alpha_{\theta}, \mathcal{W}_{\mathcal{A}}} \mathcal{L}_{\text{train}}(\mathcal{A}_{\theta}, \mathcal{W}_{\mathcal{A}})), \tag{1}$$

where a bilevel optimization manner is usually adopted to solve Eq.(1).

Most state-of-the-art differentiable NAS [6, 22, 32] methods apply a **softmax** function to calculate the magnitude of each operation and relax the discrete architecture into continue representation. A discrete architecture is obtained by applying an **argmax** function on the magnitude matrix after the supernet training. NAO [23] utilizes the LSTM based autoencoder to transform the discrete architecture into a continuous space. However, there is no injective constraint in these transformations to theoretically guarantee that the optimization in the continuous latent space is equivalent to discrete space. Chen and Hsieh [7] points out that the incongruence in most differentiable NAS is correlated with the Hessian norm of architecture parameters, which constantly increases during the supernet training[36]. Different from continuous relaxation and LSTM based autoencoder, recent works on graph neural network [33, 38] theoretically show that, the variational graph autoencoder is able to injectively transform the directed acyclic graphs (DAGs, which are used to represent architectures in NAS) into continuous representations through an asynchronous message passing scheme with a solid theoretical foundation, and guarantee the optimization in the continuous latent space is equivalent to discrete space in neural architecture search and Bayesian network learning.

After transforming the discrete architecture into continuous space, differentiable NAS conduct continuous optimization to update the continuous architecture representation $\alpha_{\theta}$ only along the

gradient of validation performance [6, 22, 23]:

$$\alpha_\theta^{i+1} \leftarrow \alpha_\theta^i - \gamma \nabla_{\alpha_\theta} \mathcal{L}_{\texttt{val}}(\alpha_\theta, \mathcal{W}^*), \tag{2}$$

where $\gamma$ is the learning rate, and $\mathcal{W}^*$ is approximated by adapting $\mathcal{W}$ using only a single training step [11, 19] with descending $\nabla_\omega \mathcal{L}_{\texttt{train}}(\mathcal{W}_{\mathcal{A}}(\alpha_\theta^i))$. This differentiable method improves the efficiency of One-Shot NAS as it obtains the most promising architecture once the supernet is trained. However, as architectures with better performance in the early stage would be trained more frequently,this method clearly entails the *rich-get-richer problem* [1, 40].

**Catastrophic Forgetting** [17, 21] usually occurs when sequentially training a model for several tasks. Given a neural network with optimal parameters $\omega^*$ on task $T_1$, the performance on $T_1$ declines dramatically after this model being trained on task $T_2$, since the weights in the networks are changed to optimize the objectives in $T_2$. Several recent works [5, 20, 35, 39] also observed the catastrophic forgetting in the One-Shot NAS, where the learning of new architecture in the supernet deteriorates the performance of previous architectures. Yu et al. [35] observed that the model with more shared weights achieves worse validation performance based on the supernet. Li et al. [20] demonstrated that KL-divergence between true parameter posterior and proxy posterior (based on weight sharing) also increases during the supernet training, making the weight sharing strategy unreliable. Benyahia et al. [5] defined it as the *multi-model forgetting*: when several models with shared parameters are applied to a single dataset $\mathcal{D}$, the learning of the current model on dataset $\mathcal{D}$ is supposed to deteriorate the performance of other previous models.

Since the multi-model forgetting deteriorates the performance of other architectures containing partially shared weights [5], the architecture parameters are supposed to move towards those areas without partially shared weights based on the gradient $\nabla_{\alpha_\theta} \mathcal{L}_{\texttt{val}}(\alpha_\theta, \mathcal{W})$ rather than those promising architectures. This catastrophic forgetting deteriorates the predictive ability of supernet and the efficiency of differentiable NAS, and several recent works [5, 20] try to overcome this catastrophic forgetting. Li et al. [20] propose to limit the number of candidate models during each step of architecture search to reduce the KL-divergence, through adopting a progressive search space shrinking strategy that only searches for a partial model in each step. Zhang et al. [39] utilizes the replay-buffer paradigm to overcoming forgetting in One-Shot NAS, by selecting the most representative architectures to regularize the supernet training. Benyahia et al. [5] propose the WPL loss function to maximize the joint posterior probability to overcome this forgetting, through regularizing the network's parameters base on the importance of each parameter [17]. While different from EWC [17], WPL only counts the shared parameters between current architecture and one previous architecture, and it could be seen as a variant of online EWC [31].

## 3 Methodology

Our novel approach consists of two key components. First, we develop an exploration enhancement module to overcome the *rich-get-richer* problem in a differentiable space. Second, we develop a architecture complementation loss function for relieving *catastrophic forgetting*. More details follow.

### 3.1 Exploration Enhancement in the Differentiable Space

**Differentiable latent space transformation** As mentioned before, existing differentiable NAS methods usually adopt a simple continuous relaxation [22] to transform the discrete neural architectures (usually represented as DAGs) into a continuous space. As they could hardly guarantee that this transformation is injective, they suffer the problem of *incongruence*. For this problem, we adopt an asynchronous message passing scheme based graph neural network (GNN) to encode the neural architecture into an injective space. Different from encoding the graph in many GNNs, we encode the computation $C$ that is the final output of the neural network into a continuous representation $\mathbf{z}$. A function $\mathcal{U}$ is utilized to update the hidden state of each node based on its neighbors' hidden states and its vertex type: $\mathbf{h}_v = \mathcal{U}(\mathbf{x}_v, \mathbf{h}_v^{\text{in}})$, where $\mathbf{h}_v^{\text{in}}$ is obtained by aggregating its all predecessors, $\mathbf{h}_v^{\text{in}} = \mathcal{G}(\{\mathbf{h}_u : u \rightarrow v\})$, and $\mathcal{G}$ is an aggregation function. According to the theory of graph neural networks [33] and in [38], if the aggregation function $\mathcal{G}$ is invariant to the order of its inputs, then the computation encoder is permutation-invariant. Furthermore, we know that the graph encoder maps $C$ to $\mathbf{z}$ injectively if the aggregation function $\mathcal{G}$ and the updating function $\mathcal{U}$ in the graph encoder are injective.

The above injectiveness indicates that, there is a one-to-one mapping from latent representation to an architecture with a graph autoencoder, and vice versa, and we could equivalently conduct differentiable optimization on the latent continuous space for architecture search. In our graph decoder, an MLP is first applied to the latent vector $\mathbf{z}$ to obtain the initial hidden state $\mathbf{h_0}$ which is fed to $\text{GRU}_d$. Then the decoder constructs a DAG node by node based on the existing graph's state. The detailed implementation of the variational graph autoencoder based on [38] for our differentiable NAS could be found in the Appendix **A**.

**Exploration Enhancement** After transforming the discrete architecture into continuous space, existing differentiable NAS methods all conduct continuous optimization to update the continuous architecture representation $\alpha_\theta$ only along the gradient of validation performance based on Eq.2. Such a method would easily get into the *rich-get-richer problem*. To overcome this problem, we add the novelty into the gradient to enhance exploration to avoid local optima in architecture search, and update the architecture according:

$$\alpha_\theta^{i+1} \leftarrow \alpha_\theta^i - (1-\gamma)\nabla_{\alpha_\theta^i}\mathcal{L}_{\texttt{val}}(\alpha_\theta^i, \mathcal{W}^*) - \gamma\nabla_{\alpha_\theta^i}N(\alpha_\theta^i, A), \tag{3}$$

where $N(\alpha_\theta^i, A)$ is to measure the novelty of architecture $\alpha_i$ from the archive $A$ (which contains $N$ previously visited architectures). After architecture update, the promising architecture with continuous representation $\alpha_\theta^{i+1}$ is then fed to the graph decoder to obtain the discrete architecture $\alpha_{i+1}$, and the weights $\omega = \mathcal{W}_A(\alpha_{i+1})$ in the supernet are updated by descending $\nabla_\omega\mathcal{L}_{\texttt{train}}(\mathcal{W}_A(\alpha_{i+1}))$. While it is intractable to measure the novelty of architectures in the discrete space, we could calculate the probability density function of $\alpha_\theta^i$ drawn from the distribution formulated by continuous architectures $\alpha_\theta$ in the archive $A$, which is also called as probabilistic novelty detection in latent space.

We describe the trained graph encoder $E$ as a mapping $E : \mathbb{R}^m \to \mathbb{R}^n$, $m > n$, and decoder $D$ as mapping $D : \mathbb{R}^n \to \mathbb{R}^m$ that defines a parameterized manifold of dimension $n$, $\mathcal{M} \equiv D(\mathbb{R}^n)$, and every architecture $\alpha_i$ could be sampled with noise $\xi_i$ through $\alpha_i = D(\alpha_\theta^i) + \xi_i$, where $\alpha_\theta^i \in \mathbb{R}^n$. Assuming the decoding function $D$ is smooth enough [28, 41], and using first-order Taylor expansion at a given point $\alpha_i \in \mathbb{R}^m$, we have

$$D(\alpha_\theta) = D(\alpha_\theta^i) + J_D(\alpha_\theta^i)(\alpha_\theta - \alpha_\theta^i) + O(\|\alpha_\theta - \alpha_\theta^i\|^2), \tag{4}$$

where $J_D(\alpha_\theta^i) \in \mathbb{R}^{m \times n}$ is the Jacobi matrix of $D$ at $\alpha_\theta^i$. The tangent space of $D$ at $\alpha_\theta^i$ could be represented as $\mathcal{T}_{\alpha_\theta^i} = \texttt{span}(J_D(\alpha_\theta^i))$. Let $J_D(\alpha_\theta^i) = U^\| S V^\top$ be the singular value decomposition (SVD) of the Jacobi matrix at $\alpha_\theta^i$, we have $\mathcal{T}_{\alpha_\theta^i} = \texttt{span}(J_D(\alpha_\theta^i)) = \texttt{span}(U^\|)$ [27, 41]. After defining $U^\perp$ as the orthogonal compliment of $U^\|$ that $U = [U^\| U^\perp]$ is a unitary matrix, we could represent the data point $\alpha_i$ in the rotated coordinates:

$$w = U^\top \cdot \alpha_i = \begin{bmatrix} U^{\|\top} \cdot \alpha_i \\ U^{\perp\top} \cdot \alpha_i \end{bmatrix} = \begin{bmatrix} w^\| \\ w^\perp \end{bmatrix}, \tag{5}$$

where the component $w^\|$ is parallel to $\mathcal{T}$, and $w^\perp$ is orthogonal to $\mathcal{T}$ as the noise $\xi$.

**Lemma 1** *Suppose we have a decoder $D$ with its tangent space represented as $\mathcal{T}$. Given a random variable $A$ formed by a set of architectures, the random variable $W$ is obtained from $A$ after coordinates rotation $W = U^\top \cdot A$, which contains two parts: $W^\|$ that is parallel to $\mathcal{T}$, and $W^\perp$ that is orthogonal to $\mathcal{T}$. Defining $p_A(\alpha_i)$ as the probability density function describing $\alpha_i$ drawn from $A$, we have $p_A(\alpha_i) = p_W(w)$, and*

$$p_A(\alpha_i) = p_W(w) = p_W(w^\|, w^\perp) = p_{W^\|}(w^\|)p_{W^\perp}(w^\perp)$$

$$\approx \left|\texttt{det}S^{-1}\right| p_{A_\theta}(\alpha_\theta) \cdot \frac{\Gamma(\frac{m-n}{2})}{2\pi^{\frac{m-n-1}{2}}\|\bar{w}^\perp\|^{m-n-1}}p_{\|W^\perp\|}(\|\bar{w}^\perp\|). \tag{6}$$

**Proof** Provided in the Appendix **B**. $\square$

Based on Lemma 1, we could calculate the novelty of a new sampled architecture $\alpha_j$ from $A$, through measuring the probability density function that the new sampled architecture located in the distribution formed by the archive $A$: $N(\alpha_j) = -\textbf{log}(p_A(\alpha_j))$. Our enhancement module encourages searching a novel architecture instead of always sampling well-trained architectures in previous rounds, avoiding the local optimal.

**Algorithm 1** E²NAS

---

1: **Input**: Trained encoder $E$ and decoder $D$, training dataset $\mathcal{D}_{train}$ and validation dataset $\mathcal{D}_{valid}$.
2: Initial architecture archive $A = \emptyset$;
3: Randomly initialize architecture parameter $\alpha_\theta$ and supernet weights $\mathcal{W}_{\mathcal{A}}(\alpha)$;
4: **while** *not done* **do**
5:     Sample batch of $\mathcal{D}_{train}$, decode $\alpha_\theta$ to get $\alpha$ based on $D$, get the complementary architecture $\alpha^c$, and update the supernet weights $\mathcal{W}_{\mathcal{A}}(\alpha)$ based on Eq. (7), and add architecture $\alpha$ into $A$;
6:     Sample batch of $\mathcal{D}_{valid}$, and update $\alpha_\theta$ based on Eq. (3);
7: **end while**
8: Decode $\alpha_\theta$ to obtain the best $\alpha^*$ based on the trained decoder $D$.
9: Retrain $\alpha^*$ and get the best performance on the test dataset $\mathcal{D}_{test}$.
10: **Return**: architecture $\alpha^*$ with best performance.

---

## 3.2 Overcoming Multi-Model Forgetting through Architecture Complementation

As described in Section 2, the differentiable NAS is built upon One-Shot NAS, which trains numerous architectures with partially shared weights on a single dataset. Without losing generality, this paper also considers the typical scenario that only one architecture (a single path) in the supernet is trained in each step of architecture search. Now we simply define each step of supernet training, $\text{argmin}\mathcal{L}_{\text{train}}(\alpha_\theta^i, \mathcal{W}_{\mathcal{A}}) = \text{argmin}\mathcal{L}_{\text{train}}(\mathcal{W}_{\mathcal{A}}(\alpha_i))$, as a task, and the supernet is trained on multiple sequential tasks through a lifelong learning setting [8, 30] or a online multi-task learning setting [10]. The catastrophic forgetting is an inevitable problem in the two scenarios and also differentiable NAS. The supernet in differentiable NAS is unable to accumulate the newly learned knowledge in a manner consistent with the past experience, and usually forgets the past learned tasks when it is trained on a new task. This phenomenon is termed as *multi-model forgetting* in [5].

To mitigate this multi-model forgetting, a mainstream approach is to select several representative tasks from a recent buffer for the replay or soft regularization [16]. The selection strategy of replay task should not be limited to the most recently experienced tasks, and it should also maximize the diversity of tasks in the replay buffer [3, 39], to balance the stability and plasticity [25]. In this paper, we select not only the last architecture $\alpha_{i-1}$ into the replay buffer, but also another *complementary architecture* $\alpha_i^c$ that is orthogonal to $\alpha_{i-1}$ to maximize the diversity of the replay buffer (we only select two architectures into the replay buffer for efficiency). Fig.3 in the Appendix shows how to select complementary architecture in our method, which makes the two following conditions hold true: $\omega_i \cap \{\omega_{i-1} \cup \omega_i^c\} = \omega_i$, and $\omega_{i-1} \cap \omega_i^c = \emptyset$, where $\omega_i = \mathcal{W}_{\mathcal{A}}(\alpha_i)$ and $\omega_i^c = \mathcal{W}_{\mathcal{A}}(\alpha_i^c)$.

The loss function for the supernet training in step $i$ is now defined as Eq.(7) when we convert the two architectures in the replay buffer to a soft regularization:

$$\mathcal{L}_c(\omega_i) = (1 - \varepsilon)\mathcal{L}_2(\omega_i) + \varepsilon(\mathcal{L}_2(\omega_i^c) + \mathcal{L}_2(\omega_{i-1})) + \eta\mathcal{R}(\omega_i), \tag{7}$$

where $\mathcal{L}_2(\omega_i)$ is the cross-entropy loss for architecture $\alpha_i$ in training set, and $\mathcal{R}$ is the $\ell_2$ regularization term. $\varepsilon$ is the trade-off to control the supernet training, whether to push the weights of current architecture to optimal or prevent deteriorating other architectures' performance in the supernet.

Our proposed complementation loss function in Eq.(7) is related to the WPL loss function [5] for overcoming catastrophic forgetting in One-Shot NAS, where WPL tries to maximize the joint posterior probability $p(\omega_{i-1}, \omega_i \mid \mathcal{D})$ in each step of supernet training. Different from WPL which only considers one previous architecture $\alpha_{i-1}$, this paper considers one additional complementary architecture $\alpha_i^c$ that is orthogonal to $\alpha_{i-1}$ during the supernet training. Given two posterior probability $p_1 = p(\omega_{i-1}, \omega_i \mid \mathcal{D})$ and $p_2 = p(\omega_i^c, \omega_i \mid \mathcal{D})$ on a dataset $\mathcal{D}$, we have the following Lemma.

**Lemma 2** *Given previous architectures $\alpha_{i-1}$ with parameters $\omega_{i-1}$, current architecture $\alpha_i$ with with parameters $\omega_i$, and the complementary architecture $\alpha_i^c$ with with parameters $\omega_i^c$, the proposed complementation loss function in Eq.(7) corresponds to maximize $p_1 * p_2$ in each step of supernet training of One-Shot NAS.*

Detailed proof of Lemma 2 could be found in the Appendix **C**. We could observe from Lemma 2 that the proposed loss function $\mathcal{L}_c$ is identical to the WPL loss function when we consider one more complementary architecture, and our loss function is more efficient to calculate without the need of estimating the Fisher information matrix or keeping the previous models in optimal points [5].

Table 1: Comparison results with state-of-the-art NAS approaches on NAS-Bench-201.

| Method | CIFAR-10 | | CIFAR-100 | | ImageNet-16-120 | |
|---|---|---|---|---|---|---|
| | Valid(%) | Test(%) | Valid(%) | Test(%) | Valid(%) | Test(%) |
| ENAS | 37.51±3.19 | 53.89±0.58 | 13.37±2.35 | 13.96±2.33 | 15.06±1.95 | 14.84±2.10 |
| RandomNAS* | 85.63±0.44 | 88.58±0.21 | 60.99±2.79 | 61.45±2.24 | 31.63±2.15 | 31.37±2.51 |
| DARTS (1st) | 39.77±0.00 | 54.30±0.00 | 15.03±0.00 | 15.61±0.00 | 16.43±0.00 | 16.32±0.00 |
| DARTS (2nd) | 39.77±0.00 | 54.30±0.00 | 15.03±0.00 | 15.61±0.00 | 16.43±0.00 | 16.32±0.00 |
| SETN | 84.04±0.28 | 87.64±0.00 | 58.86±0.06 | 59.05±0.24 | 33.06±0.02 | 32.52±0.21 |
| NAO* | 82.04±0.21 | 85.74±0.31 | 56.36±3.14 | 59.64±2.24 | 30.14±2.02 | 31.35±2.21 |
| GDAS* | 90.03±0.13 | 93.37±0.42 | 70.79±0.83 | 70.35±0.80 | 40.90±0.33 | 41.11±0.13 |
| $E^2$NAS | **90.94±0.83** | **93.89±0.47** | **71.83±1.84** | **72.05±1.58** | **45.44±1.24** | **45.77±1.00** |

The hyperparameters of $E^2$NAS are set as $\varepsilon$=0.5 and $\gamma$=$\mathtt{Sig}_\gamma(10)$ in this experiment. The best single run of our $E^2$NAS (with *random seed 0*) achieves 94.22%, 73.13%, and 46.48% test accuracy on CIFAR-10, CIFAR-100, and ImageNet, and the optimal performance on these datasets are 94.37%, 73.51%, and 47.31%, repectively.

In our **$E^2$NAS**, we train the encoder $E$ and decoder $D$ offline to enhance efficiency. Following most One-Shot NAS methods [11, 19], we only train a single-path architecture during each step of supernet training, and utilize the bilevel optimization to alternatively learn the architecture parameters and supernet weights. Generally, there are only two additional parameters that need to be specified, $\varepsilon$ for the supernet training and $\lambda$ for architecture parameter learning, in our **$E^2$NAS**. As a result, our approach can be implemented easily, and Algorithm 1 presents a simple implementation.

# 4 Experiments

The high computational cost of evaluating architectures is the major obstacle of analyzing and reproducing One-Shot NAS methods, and it is hard to reproduce current NAS methods under the same experimental setting for a fair comparison. Several recent works try to build benchmark datasets [13, 34, 37] to relieve this difficulty. In this section, we adopt the NAS-Bench-201 [13] as the benchmark dataset to analyze our $E^2$NAS. The search space in NAS-Bench-201 contains four nodes with five associated operations, resulting in 15625 cell candidates. Although the search space in NAS-Bench-201 is much simpler than the NAS common search space, the ground-truth test accuracy for all candidates in the search space is reported, and this benchmark dataset could greatly reduce the computational requirements in the analysis of One-Shot NAS methods with reproducible results.

## 4.1 Reproducible Comparison with Baselines

The comparison results on NAS-Bench-201 with NAS baselines are demonstrated in Table 1, where we report the statistical results from independent search experiments with different *random seeds* (The random seeds for all experiments on NAS-Bench-201 are set as {0,1}.). The peer algorithms include ENAS [26], RandomNAS [19], DARTS (1st, 2nd) [22], SETN [12], NAO [23], and GDAS [11]. It is inspiring our $E^2$NAS achieves the state-of-the-art results on all the three datasets in NAS-Bench-201, and significantly outperforms other baselines, especially in the CIFAR-100 and ImageNet datasets. Although our method only obtains limited improvements than GDAS in CIFAR-10, we need to notice that it is difficult to gain much more improvements on this dataset since the optimal performance is 94.37%. Our $E^2$NAS with *random seed 0* obtains a 94.22%, 73.13%, and 46.48% on CIFAR-10, CIFAR-100, and ImageNet, respectively, which are almost equal to the optimal point in NAS-Bench-201 dataset. These results demonstrate the effectiveness of our method, which employs an injective transformation to resolve incongruence, an exploration enhancement to avoid local optimum, and a novel architecture complementation to overcome the catastrophic forgetting. In the following, we further investigate how the proposed $E^2$NAS benefits from these three components: injective transformation, exploration enhancement and architecture complementation in differentiable One-Shot NAS.

Table 2: Analysis of E$^2$NAS with different $\gamma$ on NAS-Bench-201. The first block shows results of several differentiable NAS baselines, and the second illustrates results of our E$^2$NAS with different $\gamma$.

| Method | | CIFAR-10 | | CIFAR-100 | ImageNet-16-120 |
|---|---|---|---|---|---|
| | | Test(%) | Best(%) | Test (%) | Test (%) |
| DARTS (1st) | | 54.30±0.00 | 54.30±0.00 | 15.61±0.00 | 16.32±0.00 |
| DARTS (2st) | | 54.30±0.00 | 54.30±0.00 | 15.61±0.00 | 16.32±0.00 |
| SETN | | 87.64±0.00 | 87.64±0.00 | 59.05±0.24 | 32.52±0.21 |
| NAO* | | 85.74±0.31 | 86.39±1.31 | 59.64±2.24 | 31.35±2.21 |
| GDAS-A | | 89.73±3.33 | 93.49±0.24 | 62.15±6.86 | 35.34±4.81 |
| GDAS* | | 93.37±0.42 | 93.78±0.16 | 70.35±0.80 | 41.11±0.13 |
| E$^2$NAS | 0 | 92.75±0.56 | 93.79±0.17 | 68.75±0.35 | 43.19±2.10 |
| | 0.2 | 82.53±12.07 | 93.57±0.30 | 54.26±14.05 | 19.73±6.43 |
| | 0.5 | 93.34±0.30 | 93.85±0.09 | 70.41±0.76 | 44.43±0.90 |
| | 0.8 | 93.75±0.00 | 93.77±0.02 | 70.96±0.00 | 45.49±0.00 |
| | $\mathtt{Sig}_\gamma(1)$ | 93.82±0.15 | 93.87±0.07 | 70.52±1.01 | 46.10±0.52 |
| | $\mathtt{Sig}_\gamma(2)$ | 93.72±0.30 | 94.29±0.07 | 71.63±1.20 | 45.20±0.24 |
| | $\mathtt{Sig}_\gamma(5)$ | 93.78±0.16 | 94.19±0.19 | 70.55±0.44 | 44.97±0.72 |
| | $\mathtt{Sig}_\gamma(10)$ | 93.43±0.29 | 94.04±0.08 | 70.56±0.30 | 45.07±0.62 |

## 4.2 Analysis of Continuous Transformation

In this subsection, we conduct comparison experiments to demonstrate the effectiveness of our injective transformation, and all experiments are conducted without exploration enhancement that $\gamma$ (in Eq. 3) is set as 0, nor relieving catastrophic forgetting that $\varepsilon$ (in Eq. 7) is set as 0. As described in Section 3.1, we first adopt a graph autoencoder to injectively transform the discrete architecture into an equivalently continuous latent space, and then conduct differentiable optimization for architecture search as common differentiable NAS. Apart from the test accuracy of the searched architecture in the last iteration, we further demonstrate the test accuracy of the best searched architecture in all iterations to present the exploration ability in differentiable One-Shot NAS (Best (%) in Table 2). We first investigate the effectiveness of our proposed injective transformation method.

The first block in Table 2 contains several differentiable NAS baselines with different continuous transformation methods. The DARTS, SETN, and GDAS all adopt the common continuous relaxation, and NAO utilizes an LSTM based autoencoder to transform the discrete architecture into continuous. We need to notice that, different from DARTS, GDAS incorporates the uncertainty (exploration) into the architecture sampling during the supernet training through Gumbel-Max trick. To remove this effect, we consider a variant of GDAS, GDAS-A, which directly samples architectures through **argmax** during the supernet training. We consider GDAS-A as a baseline since it is the same as our E$^2$NAS($\gamma = 0$ and $\varepsilon = 0$), except that GDAS-A adopts the common continuous relaxation while E$^2$NAS uses the proposed continuous transformation method. We can observe from Table 2 that our E$^2$NAS ($\gamma = 0$ and $\varepsilon = 0$) without exploration enhancement and architecture complementation still outperforms all baselines with different transformation methods in terms of test accuracy and the best accuracy on the three datasets. These results reveal the effectiveness of the transformation method, which could more accurately and injectively transform the discrete architecture into continuous representation.

## 4.3 Analysis of Exploration Enhancement

In the following, we investigate how exploration enhancement affects the performance of our E$^2$NAS and the necessity of exploration in differentiable architecture search. In our E$^2$NAS, a bigger $\gamma$ enhances the exploration to avoid local optimal, and a smaller $\gamma$ guarantees better solutions with higher validation performance. In this experiment, we devise a Sigmoid function (defined as $\gamma(t)$ and described in Appendix D) to adapt the $\gamma$ during the supernet training to balance the exploration and exploitation, which is supposed to avoid local optimal in the early architecture search stage, and guarantees better solutions with higher validation performance in the later stage. We set eight different settings for $\gamma$, including four static settings and four Sigmoid-type settings, as shown in the third block of Table 2. The results of our E$^2$NAS demonstrate that enhancing exploration helps to improve

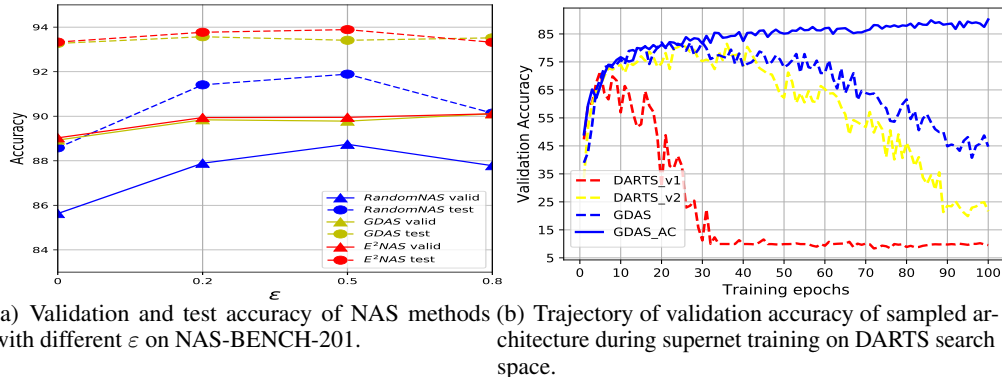

(a) Validation and test accuracy of NAS methods with different $\varepsilon$ on NAS-BENCH-201.

(b) Trajectory of validation accuracy of sampled architecture during supernet training on DARTS search space.

Figure 1: Analysis of architecture complementation on NAS-BENCH-201 dataset and DARTS search space [11, 22]. In "GDAS-AC", we replace the normal loss function during the supernet training with the proposed $\mathcal{L}_c$ defined in Eq. (7).

the performance of differentiable One-Shot NAS, where most $\gamma$ settings all improve the performance of E$^2$NAS. Our E$^2$NAS is also very robust to $\gamma$, especially for dynamic $\gamma$, where all $\mathtt{Sig}_\gamma$ achieve satisfying results. More importantly, $\gamma$ significantly enhances the exploration ability of our E$^2$NAS, and the best accuracy during the architecture search of our E$^2$NAS ($\mathtt{Sig}_\gamma(2)$) reaches 94.29$\pm$0.07%, which greatly outperform our E$^2$NAS without exploration enhancement ($\gamma = 0$). We could find that GDAS also achieves good performance in this dataset and greatly outperforms GDAS-A. One potential reason is that GDAS also introduces the exploration into the architecture search [11], which could improve the performance. Nevertheless, our method E$^2$NAS (with $\mathtt{Sig}_\gamma(10)$) still outperforms GDAS, showing impressive results.

## 4.4 Analysis of Architecture Complementation

As discussed in Section 3.2, $\varepsilon$ is the essential hyperparameter in our $\mathcal{L}_c$, and we study the impact of this hyperparameter on reliving catastrophic forgetting in the One-Shot NAS in this experiments. Apart from our E$^2$NAS (we set a fixed $\gamma$=$\mathtt{Sig}_\gamma(10)$ in this experiment), we also apply the proposed $\mathcal{L}_c$ to two popular One-Shot NAS: RandomNAS [19] and GDAS [11]. Figure 1 (a) presents the results for different One-Shot NAS methods with four $\varepsilon$ values. The results show our $\mathcal{L}_c$ could significantly improve the search results for not only our E$^2$NAS but also other NAS baselines. Compared with normal cross-entropy loss function ($\varepsilon$=0), our $\mathcal{L}_c$ could greatly enhance the search results for RandomNAS, GDAS, and our E$^2$NAS, and a medium $\varepsilon$ ($\varepsilon$=0.2 or 0.5) is recommended for all of the three methods.

Figure 1 (b) tracks the validation accuracy of the sampled architectures during the supernet training for differentiable One-Shot NAS methods on a common convolutional search space [11, 22]. We could find that the three differentiable One-Shot NAS baselines, DARTS_v1, DARTS_v2, and GDAS, all suffer from catastrophic forgetting, where the validation accuracy through inheriting weights from the supernet drops dramatically with the supernet training. The curves in Figure 1 (b) demonstrate that our proposed loss function $\mathcal{L}_c$ (as illustrated by curve GDAS_AC) could effectively relieve the catastrophic forgetting in differentiable One-Shot NAS. The performance of architectures by inheriting weights in this curve is getting better with the supernet training, making the assumption in bilevel optimization based differentiable NAS hold true.

## 5 Conclusion and Future works

This paper originally enhances the intelligent exploration of differentiable Neural Architecture Search in the latent space. A variational graph autoencoder is adopted to inject the discrete architecture space into an equivalently continuous latent space, and a probabilistic exploration enhancement method is devised to encourage the intelligent exploration during the supernet training in differentiable One-Shot NAS. We further proposed an architecture complementation loss function to relieve the catastrophic

forgetting in differentiable One-Shot NAS, and theoretically demonstrate the proposed loss function is identical to concurrent works and easier to calculate. Experimental results on a NAS benchmark dataset show the effectiveness of the proposed method. In future work, we will focus on Neural Architecture Search with Bayesian Neural Network, and also relieving the catastrophic forgetting in this case. Leveraging human knowledge in neural network design to search for architectures with better transferable ability is also one of our future directions.

## Broader Impact

Automatic Machine Learning (AutoML) aims to build a better machine learning model in a data-driven and automated manner, compensating for the lack of machine learning experts and lowering the threshold of various areas of machine learning to help all the amateurs to use machine learning without any hassle. These days, many companies, like Google and Facebook, are using AutoML to build machine learning models for handling different businesses automatically. They especially leverage the AutoML to automatically build Deep Neural Networks for solving various tasks, including computer vision, natural language processing, autonomous driving, and so on. AutoML is an up-and-coming tool to take advantage of the extracted data to find the solutions automatically.

This paper focuses on the Neural Architecture Search (NAS) of AutoML, and it is the first attempt to enhance the intelligent exploration of differentiable One-Shot NAS in the latent space. The experimental results demonstrate the importance of introducing uncertainty into neural architecture search, and point out a promising research direction in the NAS community.

It is worth notice that NAS is in its infancy, and it is still very challenging to use it to complete automation of a specific business function like marketing analytics, customer behavior, or other customer analytics.

## Acknowledgments and Disclosure of Funding

This work was supported in part by Australian Research Council (ARC) Discovery Early Career Researcher Award (DECRA) under grant no. DE190100626, Air Force Research Laboratory and DARPA under agreement number FA8750- 19-2-0501.

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
