[Supplementary Material]

# APPENDIX:

In this section, we provide the details of our implementation and proofs for reproducibility [2].

## A. Implementation of the Variational Graph Autoencoder in Differentiable NAS

Same as Zhang et al. [36], we also use the gated sum and gated recurrent unit (GRU) as the aggregate and update function, respectively.

$$\mathbf{h}_v^{\text{in}} = \sum_{u \to v} g(\text{Concat}(\mathbf{h}_u, \mathbf{x}_{\text{uid}})) \odot m(\text{Concat}(\mathbf{h}_u, \mathbf{x}_{\text{uid}}))$$

$$\mathbf{h}_v = \text{GRU}_e(\mathbf{x}_v, \mathbf{h}_v^{\text{in}}). \tag{8}$$

This encoding method also makes the framework similar to traditional RNNs for sequences.

After the optimization in the latent continuous space, we now need to decode the latent vector $\mathbf{z}$ to a neural architecture. Following Zhang et al. [36], an MLP is first applied to the latent vector $\mathbf{z}$ to obtain the initial hidden state $\mathbf{h_0}$ which is fed to $\text{GRU}_d$. Then the decoder constructs a DAG node by node based on the existing graph's state.

- Compute $v_i$'s type distribution using an MLP $f_{add\_vetex}$ based on current graph state $\mathbf{h}_{\text{G}} := \mathbf{h}_{v_{i-1}}$
- Sample $v_i$'s type.
- Update $v_i$'s hidden state by $\mathbf{h}_{v_i} = \text{GRU}_d(\mathbf{x}_{v_i}, \mathbf{h}_{v_i}^{\text{in}})$
- For $j = i - 1, i - 2, ..., 1$: compute the edge probability of $(v_j, v_i)$;(b) sample the edge; (c) if a new edge is added, update $\mathbf{h}_{v_i}$ using step 3.

Zhang et al. [36] iteratively applies the above steps to generate new nodes until Step 2 samples the ending type. However, if we want to sample the same number of nodes for each neural network, we could sample the same number of nodes before the end of the iteration, and all nodes without output edge will be connected with the extra output node.

## B. Proof of Lemma 1

**Proof** Lemma 1 is under the assumption the noise $\xi$ is orthogonal to $\mathcal{T}$ and statistically independent from the manifold. Given a new data point $\bar{\alpha}$, rather than directly calculating the probability that the new points located in the distribution of the random variable $A$, we could calculate the probability of $\bar{w}$ drawn from random variable $W$ that $W$ is obtained from $A$ after coordinates rotation $W = U^\top \cdot A$, and we have

$$p_A(\bar{\alpha}) = p_W(\bar{w}) = p_{W^\parallel}(\bar{w}^\parallel) p_{W^\perp}(\bar{w}^\perp). \tag{9}$$

Firstly, we have

$$\bar{w}^\parallel = U^{\parallel^\top} \bar{\alpha} = U^{\parallel^\top}(\bar{\alpha} - \bar{\alpha}^\parallel) + U^{\parallel^\top} \bar{\alpha}^\parallel = U^{\parallel^\top} \bar{\alpha}^\parallel, \tag{10}$$

when we assume the noise $\xi$ orthogonal to $\mathcal{M}$, and have $U^\parallel(\bar{\alpha} - \bar{\alpha}^\parallel) \approx 0$. Based on Eq.(4) and Eq.(10), we have

$$\bar{w}^\parallel = U^{\parallel^\top} D(\bar{\alpha_\theta}) + U^{\parallel^\top} U^\parallel SV^\top(\alpha_\theta - \bar{\alpha_\theta}) + O(\|\alpha_\theta - \bar{\alpha_\theta}\|^2)$$

$$= U^{\parallel^\top} D(\bar{\alpha_\theta}) + SV^\top(\alpha_\theta - \bar{\alpha_\theta}) + O(\|\alpha_\theta - \bar{\alpha_\theta}\|^2). \tag{11}$$

Then $p_{W^\parallel}(\bar{w}^\parallel) = p_{A_\theta}(U^{\parallel^\top} D(\bar{\alpha_\theta}) + SV^\top(\alpha_\theta - \bar{\alpha_\theta}))$. Based on the linear transformation of probability density, we have

$$p_{W^\parallel}(\bar{w}^\parallel) = \left| \det S^{-1} \right| p_{A_\theta}(\alpha_\theta), \tag{12}$$

since V is a unitary matrix.

Then we need to calculate the second part of Eq.(6), which is also considered as noise part $\xi$. We approximate it with its average over hypersphere $\mathcal{S}^{m-n-1}$ of radius $\|w^\perp\|$, and assume that the noise with given intensity is equally present in every direction.

As $p_{W^\perp}(\bar{w}^\perp)$ is in $m - n$ dimensional Euclidean space, we could approximate it with its average over hypersphere $\mathcal{S}^{m-n-1}$ of radius $\|w^\perp\|$, where the noise with given intensity will be equally present in every direction. We should have

$$\int \frac{2\pi^{\frac{m-n-1}{2}}}{\Gamma(\frac{m-n}{2})} \left\|\bar{w}^\perp\right\|^{m-n-1} p_{W^\perp}(\bar{w}^\perp)d(\|\bar{w}^\perp\|) = 1, \tag{13}$$

and also

$$\int p_{\|W^\perp\|}(\|\bar{w}^\perp\|)d(\|\bar{w}^\perp\|) = 1. \tag{14}$$

We could let

$$p_{\|W^\perp\|}(\|\bar{w}^\perp\|) = \frac{2\pi^{\frac{m-n-1}{2}}}{\Gamma(\frac{m-n}{2})} \left\|\bar{w}^\perp\right\|^{m-n-1} p_{W^\perp}(\bar{w}^\perp), \tag{15}$$

where $\Gamma(\cdot)$ is the gamma function. So,

$$p_{W^\perp}(\bar{w}^\perp) = \frac{\Gamma(\frac{m-n}{2})}{2\pi^{\frac{m-n}{2}} \|\bar{w}^\perp\|^{m-n-1}} p_{\|W^\perp\|}(\|\bar{w}^\perp\|). \tag{16}$$

Therefor Eq.(6) is proved.

We define $N(\bar{\alpha})$ as the novelty measurement, and the lower probability that the new point locates in the distribution, the higher novelty that the new point has. And $N(\bar{\alpha})$ could be rephrased as:

$$N(\bar{\alpha}) = -\mathbf{log}(p_A(\bar{\alpha})) = -\mathbf{log}(\left|\det S^{-1}\right| p_{A_\theta}(\alpha_\theta)$$
$$\cdot \frac{2\pi^{\frac{m-n-1}{2}}}{\Gamma(\frac{m-n}{2})} \left\|\bar{w}^\perp\right\|^{m-n-1} p_{\|W^\perp\|}(\|\bar{w}^\perp\|)). \tag{17}$$

$\square$

## C. Proof of Lemma 2

**Proof** To overcoming the multi-model forgetting during the supernet training, WPL [5] regularizes the learning of current architecture by maximizing the $p(\theta_v, \theta_i \mid \mathcal{D})$. Different from WPL, we consider one more complementary architecture, $\alpha_i^c$ with weights $\theta_i^c$, and then we need to maximize two posterior probabilities as $p_1 * p_2 = p(\theta_{i-1}, \theta_i \mid \mathcal{D}) * p(\theta_i^c, \theta_i \mid \mathcal{D})$ in each step of supernet training. Now we need to prove the proposed complementation loss function in Eq.(7) corresponds to maximize $p_1 * p_2$.

Similar to WPL [5], we depict the shared weights between $\theta_i^c$ and $\theta_i$ as $\theta_s^c$, and the private weights for the two architecture are defined as $\theta_c^p$ and $\theta_i^c$. We also depict the shared weights between $\theta_{i-1}$ and $\theta_i$ as $\theta_s^{i-1}$, and the private weights for the two architecture are defined as $\theta_{i-1}^p$ and $\theta_i^{i-1}$, and we have $\theta_i \cap \{\theta_{i-1}, \theta_i^c\} = \theta_i$, and $\theta_{i-1} \cap \theta_i^c = \emptyset$. Using the Bayes' theorem, we have:

$$p_1 * p_2 = p(\theta_{i-1}, \theta_i \mid \mathcal{D}) * p(\theta_i^c, \theta_i \mid \mathcal{D}) = p(\theta_{i-1}^p, \theta_i^{i-1}, \theta_s^{i-1} \mid \mathcal{D}) * p(\theta_c^p, \theta_i^c, \theta_s^c \mid \mathcal{D})$$

$$= \frac{p(\theta_{i-1}^p \mid \theta_i^{i-1}, \theta_s^{i-1}, \mathcal{D})p(\theta_i^{i-1}, \theta_s^{i-1}, \mathcal{D})}{p(\mathcal{D})} * \frac{p(\theta_c^p \mid \theta_i^c, \theta_s^c, \mathcal{D})p(\theta_i^c, \theta_s^c, \mathcal{D})}{p(\mathcal{D})}$$

$$= \frac{p(\theta_{i-1}^p \mid \theta_s^{i-1}, \mathcal{D})p(\theta_i^{i-1}, \theta_s^{i-1}, \mathcal{D})}{p(\mathcal{D})} * \frac{p(\theta_c^p \mid \theta_s^c, \mathcal{D})p(\theta_i^c, \theta_s^c, \mathcal{D})}{p(\mathcal{D})}$$

$$\propto \frac{p(\theta_{i-1}^p, \theta_s^{i-1}, \mathcal{D})p(\theta_i^{i-1}, \theta_s^{i-1}, \mathcal{D})}{p(\theta_s^{i-1}, \mathcal{D})} * \frac{p(\theta_c^p, \theta_s^c, \mathcal{D})p(\theta_i^c, \theta_s^c, \mathcal{D})}{p(\theta_s^c, \mathcal{D})} \tag{18}$$

$$= \frac{p(\theta_{i-1}, \mathcal{D})p(\theta_i, \mathcal{D})}{p(\theta_s^{i-1}, \mathcal{D})} * \frac{p(\theta_i^c, \mathcal{D})p(\theta_i, \mathcal{D}))}{p(\theta_s^c, \mathcal{D})} = \frac{p(\theta_{i-1}, \mathcal{D})p(\theta_i, \mathcal{D})p(\theta_i^c, \mathcal{D})p(\theta_i, \mathcal{D}))}{p(\theta_s^{i-1}, \mathcal{D})p(\theta_s^c, \mathcal{D})}$$

$$= \frac{p(\theta_{i-1}, \mathcal{D})p(\theta_i, \mathcal{D})p(\theta_i^c, \mathcal{D})p(\theta_i, \mathcal{D}))}{p(\theta_i, \mathcal{D})} = p(\theta_{i-1})p(\mathcal{D} \mid \theta_{i-1})p(\theta_i^c)p(\mathcal{D} \mid \theta_i^c)p(\theta_i)p(\mathcal{D} \mid \theta_i)$$

$$= p(\theta_i)^2 p(\mathcal{D} \mid \theta_i^c)p(\mathcal{D} \mid \theta_{i-1})p(\mathcal{D} \mid \theta_i),$$

where we use the conditional independence assumption $p(\theta_1 \mid \theta_2, \theta_s, \mathcal{D}) = p(\theta_1 \mid \theta_s, \mathcal{D})$ as we train different architecture independently in line 3; We also presupposes that the parameters $(\theta_s^{i-1}, \theta_s^c)$ are independent as the same as WPL [5] in line 6, and $\{\theta_s^{i-1}, \theta_s^c\} = \theta_i$ as $\theta_i \cap \{\theta_{i-1}, \theta_i^c\} = \theta_i$; $p(\theta_{i-1})p(\theta_i^c) = p(\theta_i)$ in the line 7 since only weights $\theta_i$ of architecture $\alpha^i$ is trained in step $i$.

From Eq.(18), we could directly derive the loss function, which is the same as Eq.(7). Therefore Lemma 2 is proved. $\square$

## D. Sigmoid-Type Function for $\gamma$

In Section 4.3, we devise a Sigmoid function to adapt the $\gamma$ during the supernet training, which is defined as:

$$\gamma(t) = 1 - \texttt{Sigmoid}\Big(\big(\frac{t}{\text{total epochs}} * 2 - 1\big) * b\big),  \tag{19}$$

where $\texttt{Sigmoid}(x) = \frac{1}{1+e^{-x}}$, and we define $\texttt{Sig}_\gamma(b)$ as the $\gamma$ is scheduled based on Eq.(19) with $b$. In this way, our architecture search is supposed to avoid local optimal in the early stage, and guarantees better solutions with higher validation performance in the later stage.

Figure 2: Sigmoid-type function for the hyperparameter $\gamma$ with the training epochs based on Eq.(19).

## E. Complementary Architecture Selection

Section 3.2 theoretically demonstrates the benefit of the proposed architecture complementation loss function, and the experimental results in Section 4.4 also verify the effectiveness of our approach, which could effectively relieve multi-model forgetting in One-Shot NAS. Figure 3 gives an example of our architecture complementation. We consider a cell structure, where node 0 is the input node, node 1 and 2 are operation nodes, and node 3 is the output node which concatenates the outputs of all input and operation nodes as the output of the cell. When the input of operation node of $\alpha_i$ is same as $\alpha_{i-1}$ (take node 1 as example), $\alpha_i^c$ randomly select a different operation. While when the input of operation node of $\alpha_i$ is different from $\alpha_{i-1}$ (take node 2 as example), $\alpha_i^c$ select a same operation as $\alpha_i$. In this way, we have $\theta_i \cap \{\theta_{i-1}, \theta_i^c\} = \theta_i$, and $\theta_{i-1} \cap \theta_i^c = \emptyset$.

Figure 3: Example of obtaining $\alpha_i^c$ through our architecture complementation.

## Footnotes

[2]The codes and experimental log files on NAS-BENCH-201 dataset could be found https://github.com/MiaoZhang0525/EENAS_for_NeurIPS2020.