[Reviews · NeurIPS 2020]

Review 1

Summary and Contributions: This paper introduces E^2NAS, a differentiable neural architecture search algorithm combined with exploration enhancement and architecture complementation. The authors apply a variational graph autoencoder to map the architecture into a manifold and define the novelty of an architecture for exploration. To alleviate rich-get-richer problem, they take differentiable NAS problem as an online multi-task learning tasks and define a complementation loss function. The experiment results show the efficiency of the proposed method.

Strengths: 1. Even though the idea of autoencoder framework has already been used in NAO, they consider the probability density function on the manifold, which is novel in NAS problem. 2. They consider one-shot NAS as a multi-modeling task and convert two architectures together. 3. The experiment results are sufficient.

Weaknesses: 1. As for exploration, there are some other traditional methods like Bayesian Optimization. I wonder what the difference between BO and your way is. 2. You add a lot of hyperparameters in the framework. Is it difficult for the model to tune the hyperparameters? 3. There are some SOTA one-shot NAS methods [1, 2] missed. Compared with their reported results, your results are not the best. For example, [1] reach 46.34+/- 0 for ImageNet on NasBench-201 while you only reach 45.77. And your results on DARTS search space is not the SOTA. 4. Some Typos. For example, in line 279 “and and”.

Correctness: The claims and method are correct and the empirical methodology is correct.

Clarity: The paper is well written.

Relation to Prior Work: It is clearly discussed how this work differs from previous contributions.

Reproducibility: Yes

Additional Feedback: [Post rebuttal] I keep my original positive score since the rebuttal convinces me in some degrees.


Review 2

Summary and Contributions: The paper proposes an exploration enhancing neural architecture search with architecture complementation (E2NAS) to address several limitations in existing differentiable NAS approaches. More particularly, the paper: -Improves the theoretical foundation of equivalently perform optimization in the continuous latency space vs. in a discrete space. -Tackles the rich-get-richer problem using a probabilistic exploration enhancement method to enhance intelligent exploration during the search. -Proposes an architecture complementation based continual learning method for the supernet training, in order to force the supernet to keep the memory of previously visited architectures. -Presents a thorough evaluation on NASBench.

Strengths: The paper is well motivated and explains the limitations in existing NAS approaches.

Weaknesses: The paper is not very novel or significant in its contribution. It compiles two regularization methods to mitigate two long-standing problems in differentiable NAS, however, the proposed methods are not very novel. NAS-Bench is not a very well established benchmark that not many people are very familiar with. It is not fair to compare with existing work on NAS-bench, as most of them were not optimized on NAS-Bench. For instance, the DARTS work may work equally well with proper hyperparameter tuning and regularization. With the existing DARTS hyperparmeters, search on NAS-bench converges to networks with only identity/skip operation. In addition, NAS-bench has its limitation such that it might not reflect the realistic settings. The reviewer would like to see a stronger ImageNet result based on either transfer learning or a direct search on ImageNet. The catastrophic forgetting problem can be a more significant issue for a multi-task learning problem. However, here, even though different network architectures are sampled sequentially, you should not treat it as a multi-task learning problem. The reviewer agrees that it is a multi-model optimization problem, not a multi-task problem. The proposed methods can be summarized in a more concise and easy-to-understand language. The first method to enhance exploration is a diversity regularization method, while the second method to mitigate forgetting is a soft regularization method with a complementation loss function. In the second method, the selection of three models seems very arbitrary and not intuitive.

Correctness: Partially correct. The paper directly reports the numbers of related work (DARTS results in an example) copied from previous paper, which seems to be not right. The DARTS results were not optimized with proper hyperparmeter tuning. The catastrophic forgetting is not accurate here.

Clarity: Yes. The writing quality is ok. The related work section is detailed and thorough.

Relation to Prior Work: The method mitigating catastrophic forgetting is not new and a similar approach was proposed by the EWC work [1]. [1] Overcoming catastrophic forgetting in neural networks, https://arxiv.org/pdf/1612.00796.pdf

Reproducibility: Yes

Additional Feedback: Post-rebuttal: The rebuttal addressed some of the concerns from the reviewer. The reviewer agrees the paper tackles some fundamental limitations of differentiable search and the contribution itself can be significant. However, the reviewer still recommend evaluating the work on a more impactful dataset such as ImageNet. Evaluating on a real workload for this current paper is critical is because the paper is built upon existing approach and the two regularization techniques invented are incremental to the reviewer's opinions. The reviewer personally enjoyed reading the DARTS paper and would consider DARTS much more innovative and more ground breaking. Also, a lot of great work were optimized for real workloads like CIFAR10, ImageNet, etc.; those work are not optimized anyway for NASBench (which is only recently getting traction). Comparison with great work in an entirely different setting is unfair and should not be encouraged. The reviewer will raise the score to acknowledge the paper's contribution and the detailed rebuttal.


Review 3

Summary and Contributions: This paper addresses the deteriorated validation performance subnets with shared weights by using variational graph autoencoder to objectively transform discrete discrete architectures into the continuous space and then tackles the catastrophic forgetting problem by a exploration enhancement way in the differentiable space through architecture complementation. The author shows some theoretical understanding on optimizing architecture search in the latent continuous space is equivalent to the discrete space and extensive empirical ablation study is provided. Overall I value this paper of good technical quality.

Strengths: This paper first analyze why traditional GD-based one-shot search fails by stating the optimizing architectures in the discrete space is not very efficient and easily lead to local minimum. It then transforms the architecture latent space into continuous by using VGAE and devise a exploration enhancement with theoretical analysis. To overcome the catastrophic forgetting problem, they further propose architecture complementation as an effective regularization during the search process. The analysis of architecture complementation in Figure 1 looks technically reasonable to me. Overall I think this work advocates a new direction that by encoding architectures in the low dimensional latent space, architectures with similar structures could be grouped together, and optimizing in such smoothing-changing performance surface makes the exploitation and exploration much easier.

Weaknesses: The experiments could be further enriched by verifying the effectiveness of the proposed method on more one-shot search spaces such as NAS-Bench-1Shot1. There is a recent and concurrent work on studying the network structure of neural architectures. it shares some similar idea that by transforming neural networks in to low-dimensional spaces using GNNs, it can provide better predictive performance of the latent representations and further improve the sampling efficiency across different tasks and datasets. I would suggest authors take a look. [1] Graph Structure of Neural Networks. ICML 2020.

Correctness: The claims and method sounds correct to me. The empirical evaluations are correct.

Clarity: Yes. It is well written and I enjoy reading the work.

Relation to Prior Work: There is no too much work on this direction. NAO is one of the eariest work optimizing neural architecture in the continuous latent space but it uses LSTM as predictor and optimized in a supervised manner. There are some concurrent work on evaluating the effectiveness of neural architecture encodings/representation learning on either discrete/continuous space. I would suggest authors to have a read. [2] A Study on Encodings for Neural Architecture Search. arXiv:2007.04965 [3] Does Unsupervised Architecture Representation Learning Help Neural Architecture Search?arXiv:2006.06936 [4] Are Labels Necessary for Neural Architecture Search? ECCV 2020.

Reproducibility: Yes

Additional Feedback: Take a look on the NAS-Bench-1Shot1 paper and [1,2,3,4] which discusses lots of useful information that coincidence with the observations in your paper.


Review 4

Summary and Contributions: This paper addresses three important problems in one-shot gradient-based NAS, namely 1) the incongruence between the continuous search space and the discrete result architecture, 2) the rich-get-richer problem where the initially good connections will be selected and reinforced during the search, and 3) the multi-model catastrophic forgetting problem when the weights are shared and trained in a one-shot manner. The authors tackle the first problem by introducing a one-to-one mapping from the latent representation to an architecture with a graph autoencoder, while introducing a replay buffer with novel losses (i.e., the loss to encourage selecting novel architectures and the loss considering the complementary architecture) to address the second and the third problems. The experiments on NAS-Bench-201, CIFAR-10/100, ImageNet-16-120 demonstrate the promising performance of the proposed method.

Strengths: 1. Three important problems in one-shot gradient-based NAS are addressed. 2. Detailed experiments and ablations. 3. This paper is well-written and easy to follow.

Weaknesses: 1. In Table 2, why gamma = 0.2 performs the worst and introduces huge variance? Should the performance change smoothly with gamma? 2. What is the definition of the complementary/orthogonal architecture? Should the union of alpha_{i-1} and alpha_i^c be the whole search space, or it just needs the union of alpha_{i-1} and alpha_i^c includes alpha_i? 3. Please consider redrawing Fig. 1 as the current version possess unnatural skewing. 4. It should be minus in Eqs. (2) and (3) for gradient descend.

Correctness: Yes, except a typo in Eqs. (2) and (3) (Please see Point 4 in the Weakness Section)

Clarity: Yes.

Relation to Prior Work: There are two recent methods dealing with the search/result space incongruence problem from other ways, please consider citing and discussing them: [1] Li et al., SGAS: Sequential Greedy Architecture Search. arXiv:1912.00195, 2019. [2] Gao et al., MTL-NAS: Task-Agnostic Neural Architecture Search towards General-Purpose Multi-Task Learning. arXiv:2003.14058, 2020.

Reproducibility: Yes

Additional Feedback: I have read the authors' rebuttal and remain my initial recommendation.

[Author Response · NeurIPS 2020]

***Response Summary.*** We truly appreciate reviewers' valuable comments and positive feedback. We are encouraged reviewers found the problems addressed by our paper are important (Reviewer 3, 4), our motivation is clear (Reviewer 1, 2, 3), and idea is novel (Reviewer 1, 3, 4). We are glad all reviewers found this paper is well written with excellent reproducibility. One primary concern was the results on DARTS search space were not SOTA. Similar as DARTS and other differentiable NAS, we also conducted more architecture search for high-performance models. We have released a new architecture on GitHub (the anonymous link in the original submission file), with 2.50% test error on CIFAR10, 15.84% on CIFAR100, and 24.37% on ImageNet, which are SOTA results under DARTS experimental settings. Thanks Reviewer 3 for the strong support, and we will provide a detailed item-wise response for other reviewers below.

*Responses to Review 1:*

**Q1**: As for exploration, there are some other traditional methods like BO. What the difference between BO and your way is. **R1**: BO needs to build probabilistic models (e.g. GP) by training multiple architectures from scratch first, while differentiable NAS only trains a supernet once. It is not intuitive to directly introduce the exploration in BO to differentiable NAS, while our exploration could be easily applied to differentiable NAS to solve rich-get-richer problem.

**Q2**: Is it difficult to tune hyperparameters? **R2**: No. There are only two hyperparameters to be tuned, and the others are default. The ablation studies in Sec 4.3, 4.4, and Appendix H show our model is robust to the two hyperparameters.

**Q3**: Some SOTA one-shot NAS methods [1,2] in the NAS-Bench-201 dataset missed. And the results on DARTS search space are not the SOTA. **R3**: The two references are missing in the review. Our best single run can achieve 46.48% for ImageNet on NAS-Bench-201 as described in the bottom of Table 1 and line 246 in the original submission, outperforming [1] (46.34% for ImageNet) provided by the reviewer. Please refer to ***Resp. Summ.*** and GitHub for the results of our new high-performance model on the DARTS search space.

*Responses to Review 2:*

**Q1**: The paper is not very novel with limited contribution, the method mitigating catastrophic forgetting is not new and similar with the previous EWC work. **R1**: This paper is the first paper introducing intelligent exploration into NAS, through the probability density function based on a graph autoencoder. We believe it is novel in NAS and Reviewer 1 and 3 also agree on it. Besides, the regularization method to mitigate forgetting in our method is totally different from EWC and WPL (EWC applied to one-shot NAS). EWC and WPL both calculate the joint posterior probability through estimating the Fisher information matrix and assuming the previous models in optimal points, while the two conditions hardly hold in differentiable NAS. We propose an architecture complementation scheme, and theoretically shows it could optimize the joint posterior probability as EWC and WPL, without the assumption of the two conditions.

**Q2**: NAS-Bench is not a well-established benchmark that not many people are very familiar with. **R2**: The NAS-Bench is a newly established benchmark with a much simpler search space, while the ground-truth test accuracy for all candidates in the search space is reported, helping the NAS methods to conduct reproducible experiments with much less computational requirements. Building a well-established benchmark is becoming a new interesting research direction in NAS, and concurrent NAS-Bench 101, NAS-Bench 201, NAS-Bench 1Shot1, et. al., all help to relieve computational requirements, and recent researches in NAS community prefer these benchmarks for enhancing reproducibility.

**Q3**: Require a stronger ImageNet result. **R3**: Please refer to ***Resp. Summ.*** and GitHub for our new SOTA results.

**Q4**: The author should not treat supernet training as a multi-task learning problem. The reviewer agrees that it is a multi-model optimization problem, not a multi-task problem. **R4**: This paper focuses on the multi-model forgetting problem in the supernet training. The multi-model forgetting in NAS is very related to catastrophic forgetting in multi-task learning, as described in Sec. 2. To avoid confusion, we have rephrased "catastrophic forgetting" to "multi-model forgetting" when describing the forgetting in NAS, and rephrased Sec.3.2.

**Q5**: In relieving forgetting, the selection of three models seems arbitrary and not intuitive. **R5**: It should be noted that our architecture complementation scheme is to select specific architectures for regularization. We theoretically show our method can optimize the joint posterior probability similar as EWC and WPL, but with less constraints. The ablation study in Appendix H shows our method outperforms other naive schemes, including WPL, random selection, and so on.

**Q6**: The paper directly compares related works copied from the previous paper without hyperparameter tuning. **R6**: The results of peer algorithms are from the original paper (NAS-Bench-201) since we adopt the same experimental settings as that paper. Furthermore, our model outperforms most peer algorithms under all hyperparameter settings.

*Responses to Review 4:*

**Q1**: In Table 2, $\gamma = 0.2$ performs the worst with huge variance? Should the performance change smoothly with $\gamma$? **R1**: The performance should change smoothly with $\gamma$. As discussed in Sec. 4.3, a dynamic $\gamma$ seems to achieve better performance, and a small and static $\gamma$ may lead to local optimal. As we conducted experiments with limited random seeds, the outliers may greatly affect the statistical results. We have conducted experiments with more random seeds and will remove outliers to obtain statistical results to avoid the effects of outliers in the final version.

**Q2**: The definition of complementary/orthogonal architecture? Should the union of $\alpha_{i-1}$ and $\alpha_i^c$ be the whole search space, or just needs the union of $\alpha_{i-1}$ and $\alpha_i^c$ includes $\alpha_i$? **R2**: We define that $\alpha_m$ is orthogonal to $\alpha_n$, so they do not share parameters $\omega_m \cap \omega_n = \emptyset$. As to the complementary architecture, since we first select the $\alpha_{i-1}$ into the replay buffer, the complementary architecture $\alpha_i^c$ is defined as $\omega_i \subseteq \{\omega_i^c \cup \omega_{i-1}\}$ that only needs the union of $\alpha_{i-1}$ and $\alpha_i^c$ includes $\alpha_i$, and $\alpha_i^c$ is also orthogonal to $\alpha_{i-1}$.

[Meta-Review · NeurIPS 2020]

The reviewers generally found this paper to be a good contribution to the NAO/NAS field, with a good motivation and strong results. There were concerns on the novelty of the work, but after considering the author’s response, particularly in relation to EWC, I think the work is sufficiently novel, especially given the relatively new domain. I would encourage the authors to include the clarifications and comparison to related work from the rebuttal in the main paper. The biggest issue that still lingers is the fact that NAS-Bench-201 is a very small benchmark. The most positive reviewer strongly encourages the authors to apply their technique to a larger benchmark such as NAS-Bench-1Shot1. The lingering concern otherwise is that it’s unclear if the approach will generalize to larger spaces, and some work to address this for the camera ready would be very beneficial.